# The G Protein-Coupled Estrogen Receptor (GPER): A Critical Therapeutic Target for Cancer

**DOI:** 10.3390/cells12202460

**Published:** 2023-10-16

**Authors:** Keith A. Hall, Edward J. Filardo

**Affiliations:** Starling Biosciences, Inc., Coralville, IA 52241, USA; keith@starlingpharmaceuticals.com

**Keywords:** GPER, ERα, ERβ, estrogen, xenoestrogens, phytoestrogens, breast cancer, metabolic disease, endocrine therapy, antiandrogen

## Abstract

Estrogens have been implicated in the pathogenesis of various cancers, with increasing concern regarding the overall rising incidence of disease and exposure to environmental estrogens. Estrogens, both endogenous and environmental, manifest their actions through intracellular and plasma membrane receptors, named ERα, ERβ, and GPER. Collectively, they act to promote a broad transcriptional response that is mediated through multiple regulatory enhancers, including estrogen response elements (EREs), serum response elements (SREs), and cyclic AMP response elements (CREs). Yet, the design and rational assignment of antiestrogen therapy for breast cancer has strictly relied upon an endogenous estrogen–ER binary rubric that does not account for environmental estrogens or GPER. New endocrine therapies have focused on the development of drugs that degrade ER via ER complex destabilization or direct enzymatic ubiquitination. However, these new approaches do not broadly treat all cancer-involved receptors, including GPER. The latter is concerning since GPER is directly associated with tumor size, distant metastases, cancer stem cell activity, and endocrine resistance, indicating the importance of targeting this receptor to achieve a more complete therapeutic response. This review focuses on the critical importance and value of GPER-targeted therapeutics as part of a more holistic approach to the treatment of estrogen-driven malignancies.

## 1. Introduction

It is an honor to contribute a feature article for this issue of *Cells* focusing on the G protein-coupled estrogen receptor (GPER), which has been linked to metaboregulation and cancer. It has been approximately twenty years since the first studies providing evidence that the orphan receptor, GPR30, promotes specific estrogen binding [1,2] and rapid, nongenomic estrogen signaling [2,3,4,5] were published. A large body of literature describes its mechanism of action and the physiological and pathophysiological responses associated with this receptor, and collectively, these findings led to its rechristening to its functional designation as GPER by the receptor nomenclature committee of the IUPHAR (International Union of Pharmacologists) in 2015 [6].

This review focuses on the need for a more holistic approach to endocrine therapy that considers the roles and interplay between all sources of estrogen, endogenous and environmental, and estrogen receptors, with a special focus on GPER, given its potential importance for cancer and metabolic regulation. This review is organized in three parts. Firstly, key arguments that support the broad family of estrogens as tumor promoters in multiple human malignancies are outlined. Secondly, the receptor-based mechanisms that coordinately promote the diverse array of intracellular signaling actions activated by estrogens are discussed. Finally, we evaluate the current modalities for endocrine therapy in the context of a broader view of estrogen action that includes environmental estrogens and GPER.

## 2. Estrogens and Their Implications in Carcinogenesis

According to the 2020 GLOBOCAN estimates, 18.1 million new cancer cases were detected worldwide, with breast cancer surpassing colon cancer as the most diagnosed cancer (2.3 million cases, 11.7%). The World Health Organization’s International Agency for Research on Cancer (IARC) tracked 685,000 breast cancer deaths during 2020 and projects that these numbers will rise to 3 million new cases and 1 million deaths by 2040 [7]. Of considerable importance is the fact that breast cancer incidence is rising rapidly in transitioning countries where mortality rates from the disease remain high. Equally alarming is the rapid rise in colon cancer, a malignancy previously recognized to have a strong genetic component, with a gender bias toward men [8,9]. While historical numbers show a striking disparity between premenopausal women and age-matched men, recent epidemiological data indicate that this difference has decreased significantly [10]. Causal factors driving colon cancer in younger adults are not clear; however, environmental factors, including diet, microbiome, and inflammation, have been proposed [11]. Equally concerning is that this trend is consistent with increasing human exposure to environmental estrogens and the rise in obesity among young adults [12]. While ER plays an important role in metabolic regulation, the link between lifestyle, metabolic disorder, and increased exposure to environmental estrogens also point to the potential significance of GPER, based on several observations: (i) many environmental estrogens exhibit high affinity for GPER [13]; (ii) the genetic silencing of GPER alters metabolic homeostasis [14]; (iii) increased exposure to environmental estrogens is associated with metabolic syndrome [15]; and (iv) metabolic syndrome is a collection of diseases, which contribute to the genesis of cancer, particularly cancers that arise from the breast [16,17] and colon [18].

Estrogens have been implicated in the pathogenesis of various cancers, including female reproductive cancer [19,20] as well as lung [21,22,23], liver [24,25,26], colon [27], and kidney cancer [28,29]. Mechanisms of carcinogenesis by estrogen center on its genotoxic or mitogenic activity on tumor target cells, and this has been reviewed in detail elsewhere [30,31]. These models suggest that receptor-based estrogen signaling promotes the expansion of the transformed population, and in some instances, the subsequent accumulation of somatic mutations occurs, which drives cancer progression via the metabolic activation of catechol estrogens or environmental estrogens, or through epigenetic mechanisms. Alternative mechanisms of carcinogenesis have focused on chronic estrogen signaling and oxidative stress [32,33]. Regardless of the precise nature of the model that drives estrogen-induced carcinogenesis, understanding the receptor-based mechanisms through which endogenous and environmental estrogens promote their biochemical and biological actions is fundamentally required.

## 3. Estrogen Promotes Biochemical Actions That Are Manifested through Two Distinct Receptor Mechanisms

Endogenous and environmental estrogens promote biochemical actions that occur over a broad range of time and span signaling events that can be measured from seconds to over several hours. These actions are manifested by three estrogen receptors, named ERα, ERβ, and GPER, which are encoded at individual genetic loci in the human genome: ERα/ESR1 (6q25.1), ERβ/ESR2 (14q23.2), and GPER (7p22.3). The molecular basis of ligand binding and intracellular signaling by each of these receptors has been reviewed in detail elsewhere [30,34,35,36,37,38], and low nanomolar binding affinities have been measured for each of these receptors for the most abundant endogenous estrogen, 17β-estradiol (17β-E2). The binding activities, including relative affinities and effective half-maximal concentrations (EC50s) of 17β-E2 and environmental estrogens for ERs or GPER, have also been widely examined, and they exhibit distinct relative binding affinities and pharmacological activities toward synthetic estrogens and environmental estrogens [6,13,39].

The biochemical signaling activities manifested by ERα, ERβ, and GPER are schematized in Figure 1. ERα and ERβ belong to the steroid hormone receptor family and act as ligand-induced transcription factors, with conserved functional domains for hormone and DNA binding. Both ERα and ERβ exhibit different ligand affinities and, in some cases, specificities for various estrogenic ligands [39]. Crystal structures for ERα and ERβ, and the allosteric changes that occur in response to ligand binding and association with molecular chaperones and accessory proteins, have been resolved and identified [40,41,42,43]. ERα and ERβ directly activate gene transcription through their ability to bind to transcriptional regulatory sites encoded in DNA, known as estrogen response elements (EREs). In contrast, GPER, a member of the G protein-coupled receptor (GPCR) superfamily, promotes its transcriptional effects indirectly from its ability to employ heterotrimeric G proteins to stimulate cAMP and EGFR signaling. The former occurs because GPER is a Gαs-linked protein [1] that directly stimulates adenylyl cyclase, which in turn converts ATP into cAMP [4]. The latter is mediated by the free Gβγ subunit dimer, which triggers the release of intracellular calcium, resulting in Src family kinase Shc-dependent proteolytic cleavage, and consequent release of membrane-tethered, heparan-bound epidermal growth factor (HB-EGF) from the exoplasmic face of the plasma membrane [44]. In this manner, estrogens act via GPER to transactivate EGFR family members [45] while in parallel stimulating cAMP-dependent signaling pathways. Although GPER actions are initiated through the activation of plasma membrane-associated enzymes (i.e., adenylyl cyclase and matrix metalloproteinases), ultimately, these pregenomic actions lead to gene transactivation [46].

Several lines of experimental evidence support the idea that GPER transcriptional activation occurs independently of ER. First, in both rodent uterine epithelial cells and human MCF-7 breast cancer cells, both active estrogens and antiestrogen (4-hydroxy-tamoxifen or ICI 182780) have been shown to be capable of promoting gene transcription via reporter plasmids containing cyclic AMP-regulated enhancer (CRE) elements [47]. Importantly, these investigations revealed that the inhibitors of RNA or protein synthesis did not block estrogen-induced cAMP signaling or CRE-mediated gene transcription, indicating that these effects did not require de novo gene expression. GPER-mediated gene transactivation was established through experiments that demonstrated c-fos gene transactivation by 17β-estradiol or phytoestrogens (quercetin or genistein) in ER-negative, GPER-positive human SKBR3 breast cancer cells or in human MDA-MB-231 cells following the reconstitution of GPER [5]. Further proof of concept demonstrating GPER-specific gene transcription was developed through additional experiments using the GPER-selective agonist, G1, and a c-fos reporter plasmid in ER-positive, GPER-positive human BG-1 ovarian carcinoma [48]. These findings are consistent with earlier work that first identified GPER as a plasma membrane receptor that can robustly activate the EGFR-to-erk-1/2 signaling conduit [3,34,45]. While the full scope of genes whose transcription is mediated through GPER-dependent EGFR transactivation has yet to be fully determined, it is well recognized that EGFR signaling can lead to the activation of transcription factors that bind to serum response elements (SREs). A broader view of gene signatures that can be traced to GPER signaling is provided by a recent study by Lappano and Maggiolini analyzing the transcriptomes of ER-negative breast cancer patients from The Cancer Genome Atlas (TCGA) and Molecular Taxonomy of Breast Cancer International Consortium (METABRIC) datasets [49]. This study showed a correlation in the expression of GPER with genes that align with pro-metastatic pathways. Similar transcriptomic analyses were performed by Xu and colleagues in a study of 360 Chinese patients with triple-negative breast cancer (TNBC), an aggressive breast cancer subtype that expresses low levels of ER, progesterone receptor (PR), and her2/neu [50]. These investigators were able to identify four distinct TNBC molecular subtypes, with the highest levels of GPER correlating with the poorest metastasis-free survival. These findings are consistent with other studies performed using clinically relevant immunohistochemical staining that have shown that GPER is expressed in the majority of TNBC cases and collectively have implications on rational drug design to combat a type of cancer traditionally thought to be nonresponsive to endocrine therapy [50,51,52]—a subject we will return to later.

## 4. Promotion of Carcinogenesis via Interplay between Estrogens and Estrogen Receptors, ER and GPER

The estrogen receptor signaling mechanisms shown in Figure 1 are at the core of carcinogenesis and can be triggered by endogenous and exogenous sources of estrogen. As such, both endogenous and exogenous estrogens have been unequivocally labeled as carcinogens [30,31]. However, an incomplete understanding of the complement of estrogen receptors expressed in cancer has led to confusion regarding the role of estrogen in carcinogenesis. Despite the strong evidentiary examples supporting a pro-oncogenic role for GPER in breast cancer, it has been suggested that GPER does not drive cancer but may act as a “tumor suppressor”, with both genetic and pharmacological data reported in support of this label. Evidence for GPER as an antioncogene is extrapolated from studies that show that the GPER promoter is hypermethylated in some tumors [53,54] and thus rendered transcriptionally inactive. It is important to point out that, in these studies, methylation was not evaluated as a global effect due to genome instability; nor was the methylation status of other genetic loci, including ER, assessed. By the criteria applied from this work, ER should also be classified as a tumor suppressor, as its promoter and locus are commonly hypermethylated in breast cancer, and because this is the primary mechanism through which ER protein expression is silenced in breast cancer [55], occurring in as many as one of three cases of breast cancer. Moreover, if ERα were defined as a tumor suppressor based on its frequent hypermethylation, this definition would contradict its role in breast carcinogenesis and the success achieved by specifically targeting this receptor with antiestrogen therapy for breast cancer. Furthermore, the assignment of the term “tumor suppressor” to GPER does not fit with retrospective clinical data showing that GPER aligns with disease progression and worse survival in breast and gynecological cancer [49,50,56,57,58,59] or with experimental data indicating that its genetic inactivation in PDX biopsies significantly reduces tumor growth [60]. Pharmacological evidence in support of the antioncogenic activity of GPER is derived from data that have shown that GPER selective agonists can promote biochemical and cellular events leading to cell death, including the disruption of tubulin [61,62], G_2_/M mitotic arrest and the upregulation of p53 and p21 [63], and the cAMP-dependent exhaustion of c-myc [64]. In the context of this argument, it is important to point out that the GPER-selective antagonist, G36, also inhibits the growth of endometrial cancer [65] and nonsmall cell lung cancer [66]. Nevertheless, further complexity is added by the fact that ER antagonists function as GPER agonists [13]. Thus, it may be more appropriate to recognize that *total estrogen exposure*—endogenous and environmental—is a chronic stimulus that promotes estrogen action, independent of receptor type and where the complementary action of these receptors is associated with estrogen-driven malignancies. This view aligns more closely with the traditional evaluation of estrogen as a carcinogenic agent (i.e., as a tumor promoter).

While the notion of estrogen carcinogenicity is clear in breast cancer, other tissues not traditionally considered as being estrogen-responsive, such as colonic epithelia, may be sensitive to estrogens, as manifested through their specific cellular receptors. This makes sense considering that human exposure to environmental estrogens can be accounted for directly in food as well as in plastic containers used to store preserved food and bottled water as well as other beverages, which ultimately reside as solid waste in the colon. While several observations support a role for estrogen in colorectal carcinogenesis (CRC) [67,68,69,70], it has been argued that the protective, or “antioncogenic”, effect of endogenous estrogen is a key reason that premenopausal women are less likely to develop colorectal cancer [69]. However, this latter theory contradicts the observation that diseases associated with disrupted gastrointestinal homeostasis (i.e., constipation and irritable bowel syndrome) are far more likely to occur in women than men [71]. Moreover, experimental evidence demonstrates that estrogen promotes profound effects upon colonic epithelia that can disrupt water and electrolyte homeostasis [72]. It is noteworthy that the best-studied pathogenic mechanism for inducing diarrhea is the enzymatic activation of the Gαs subunit protein and adenylyl cyclase following cholera toxin exposure [73], a signaling event that is also accomplished following the stimulation of GPER with 17β-E2 [4]. Nonetheless, it has alternatively been suggested that xenoestrogens that function as endocrine disruptors (e.g., bisphenol A, plant-derived polyphenols) may interfere with the antioncogenic effects of estrogen for colorectal carcinoma [70]. Because ERβ is less well expressed than ERα, it has been suggested that the balance of endogenous estrogens versus xenoestrogens may determine antioncogenesis versus pro-oncogenesis in the colon. However, it is curious that the expression of ERβ is commonly absent in advanced colorectal cancer (CRC) [74]. A more recent study has implicated GPER as a “pro-oncogenic” factor in CRC [75], and this is a research area that warrants further study.

Evidence for estrogen in the carcinogenesis of other solid neoplasms has been provided by a recent work by Ridky and colleagues that has shed some light on the “pregnancy-mediated protective effects” associated with improved patient outcomes in cutaneous melanoma [64]. This work has implicated GPER in activating cAMP-dependent gene transcription responses that result in an increased expression of melanocyte antigens, decreased expression of the oncodriver c-myc, and increased sensitivity to PD-L1 checkpoint inhibition. Similar observations by Sun and coworkers have demonstrated that the stimulation of melanoma cells with the synthetic GPER agonist, G1, promotes the cAMP-dependent activation of the microphthalmic transcription factor (MITF), which regulates tyrosinase, a rate-limiting enzyme in melanin production [76]. These results are consistent with an earlier work that identified GPER as a Gαs-coupled GPCR [1,34]. A more recent work by Ridky has shown that GPER agonism can also heighten PD-L1 sensitivity in a preclinical model of pancreatic ductal adenocarcinoma [77]. In another study, the stimulation of uveal melanoma cells with the GPER agonist LNS8801 (under development by Linnaeus Therapeutics) also induced melanocyte differentiation markers and concomitantly upregulated p53 and p21, G_2_/M mitotic arrest and the disruption of the mitotic spindle [63]. Whether or not these results are associated with cAMP-dependent GPER signaling or receptor downmodulation is unclear at present. Similarly, G2/M arrest was achieved in GPER-positive human HCC1806 TNBC cells following treatment with a GPER-targeted proteolytic targeting chimera (PROTAC), while cell cycle blockade did not occur in GPER-low human MDA-MB-231 breast cancer cells [78].

## 5. Human Exposure to Environmental Estrogens and Implications for Estrogen-Induced Carcinogenesis in the Context of Multiple Cancer-Involved Estrogen Receptors

Our environment is replete with compounds that function as estrogen mimetics known as xenoestrogens, many of which act as endocrine-disrupting chemicals (EDCs). Xenoestrogens can be readily found in food, plastics, food containers, cosmetics, and skin care products [79]. A few select examples of xenoestrogens are dietary phytoestrogens (genistein and daidzein), the synthetic environmental pollutant bisphenol A, and 4-hydoxytamoxifen. The active metabolites of the antiandrogen tamoxifen are listed in Table 1 along with relative binding affinities (RBAs) toward ERα, ERβ, and GPER.

EDCs have long been implicated in carcinogenesis [82], although their contribution to cancer has remained controversial [83]. To some extent, this confusion arises from past adherence to a receptor model that centers on the nuclear estrogen receptors ERα and ERβ but does not incorporate GPER. For example, the potential contributions of abundant xenoestrogens such as bisphenols and dietary soy isoflavones in breast cancer have been dismissed largely due to their low relative binding affinities (RBAs) for ERα, even though many of these exhibit significantly higher RBAs for GPER, and this has been previously described [6,13,39,81]. Genistein provides an example of this where the potential for it to significantly interfere with the binding of 4-hydroxytamoxifen to ER has been discounted due to the large difference in the RBAs between them: 175–257% for 4-hydroxytamoxifen versus 0.021% and 6.8% for ERα and ERβ for genistein, respectively. In contrast, the RBAs of genistein and tamoxifen to GPER are similar (2–5% and 0.3–6%, respectively), and both compounds function as GPER agonists.

However, apart from RBA values, it is equally if not more important to also consider several other factors that can potentially affect the potential carcinogenicity of EDCs, for example, the fact that the bioavailability of phytoestrogens and other EDCs relative to 17β-E2 may be significantly higher due to their inability to complex with sex hormone binding globulin (SHBG) [84]. This difference may be even higher in the context of tamoxifen, which is known to increase SHBG levels. Furthermore, this concentration difference is expected to be exacerbated in patients receiving hormone therapy such as aromatase inhibitors or ovarian suppressors to block 17β-E2 production. For patients taking tamoxifen or fulvestrant to downregulate ER, there remains the potential for GPER agonism, either from EDCs, 17β-E2, or the drug itself as tamoxifen and fulvestrant are GPER agonists (discussed in Section 9). Collectively, these factors have the potential to offset the effect of a low RBA for EDCs in terms of the potential for agonizing ER and GPER.

Daidzein is yet another example of a common dietary soy isoflavone that has importance with regard to promoting chronic estrogenic stimulation since postprandial serum concentrations of daidzein can exceed preovulatory serum concentrations of 17β-E2 by ten-fold [85], which may overcome the relatively low RBA to ER. A recent study demonstrated that daidzein-induced, GPER-dependent migration could be measured in glial cells at a half-maximal effective concentration EC_50_ of 10 nM [86]. In a separate report, *S*-(–)equol, the bacterial metabolite of daidzein, was also demonstrated to be an effective GPER agonist, with an EC_50_ of 100 nM for eNOS stimulation in endothelial cells [87]. In this context, *S*-(–)equol may be an effective physiological modulator of GPER activity since only 50% of it is serum protein-bound (50% is free and bioavailable), while <5% of 17β-E2 is not serum-bound [88]. This may have significance for breast cancer in that *S*-(–)equol producer status has been linked to lower breast cancer risk [89]; however, further studies are needed to evaluate the bacterial metabolism of soy isoflavones in breast cancer patients whose tumors contain distinct estrogen receptor profiles. In summary, while there are obviously considerable commercial drivers behind the popular usage of dietary soy isoflavones as antioxidants and anti-inflammatory agents, their potential to counteract or modulate endocrine responses remains a concern [90,91].

Increased exposure to environmental synthetic estrogens such as diethyl phthalates [92], *o,p*′-DDT [93], polychlorinated biphenyls (PCBs) [94,95], and bisphenols [96] have been hypothesized in part to contribute to the increased breast cancer risk in developed countries [97,98]. Also of concern is the synthetic estrogen, bisphenol A (BPA) which exhibits an approximate 30–60-fold lower relative binding affinity for ERα or ERβ than GPER (0.008% for ERα or ERβ compared with 0.5–1% for GPER (Table 1) [6,99]. This may have important ramifications regarding the need to target GPER, based on the sheer abundance of BPA and its easily modifiable structure. At present, more than 40 chemical analogs of BPA exist, and several of these have been reported to have high RBAs for GPER (i.e., bisphenol AF and bisphenol B) [100]. With regard to bisphenols, human exposure is significant, and as discussed below, there has been considerable attention and pressure placed upon regulatory agencies to re-evaluate current standards for public safety. This may have further importance considering the high affinity of GPER for bisphenols and the nonlinear relationship that many GPCRs demonstrate between receptor occupancy and signaling.

Bisphenols are ubiquitously utilized in the production of polycarbonate plastics, plasticized liners on packaged food, dental sealants, and thermal receipt papers [101,102]. More than 6 billion pounds of BPA alone were produced worldwide in 2020, with the US consuming more than 40% of total production [103]. The US Environmental Protection Agency (EPA) estimates that more than 1 M pounds of BPA leaches into the environment each year, and it is detected in over 95% of all people, with the highest concentrations measured in infants and children [104,105]. Their common presence in food packaging has resulted in renewed requests for the US Food and Drug Administration (FDA) to limit the use of BPA to 0.05 ng/kg of food in January 2022 [106]. While the current safe reference dose set by the FDA is at a threshold of 50 μg/kg body weight per day [96], this is significantly higher than the safety standards that have been set in Europe. In December 2021, the European Food Safety Authority (EFSA) revised its previous safety limit for BPA from 4 μg/kg body weight per day to a new standard of 0.04 ng/kg body weight per day [106]. These cautionary changes imposed by the EFSA may be justified based on multiple studies that have demonstrated that BPA and other endocrine-disrupting chemicals commonly result in nonmonotonic dose responses (NMDRs) [107], with biological/cytotoxic responses measured at very low drug concentrations that are in the range of typical human exposure levels. Several laboratory studies have provided experimental evidence demonstrating that BPA has an “instigating effect” on breast carcinogenesis [96,108]. Among the mechanistic explanations that are offered by toxicologists for nonmonotonicity is the effect of excess hormone concentration in the context of inactivated, degraded, or desensitized receptors. This explanation is particularly relevant for this review as G protein-coupled receptors (GPCRs) have been modeled as receptors to explain NMDRs [96,108].

## 6. GPER as a Therapeutic Target in Metabolic Disease and Cancer

The observation that colon and breast cancers are two malignancies that best align with central adiposity and cancer risk may further suggest a role between GPER, metabolic syndrome, and cancer. This has been borne out in part from experimental evidence demonstrating that GPER null mice develop metabolic syndrome [14] and GPER agonists effectively reduce this phenotype in mice that are fed an obesogenic diet [109]. These observations may be relevant to parallel increases in the incidence of breast cancer, obesity, and insulin resistance in socioeconomically developed countries [110]. For example, in the United States, approximately one in four adults will develop insulin resistance syndrome, which is more common in older people and will continue to increase as the population ages and becomes more obese [110]. Insulin resistance syndrome co-occurs with metabolic risk factors for type II diabetes mellitus and cardiovascular disease, including obesity, dyslipidemia, hyperglycemia, and hypertension. These core pathologies collectively define metabolic syndrome and increase the risk of developing cancer [111] (Figure 2). It may therefore not be merely coincidental that a link can be drawn between exposure to environmental estrogens and the development of metabolic syndrome [12,112]. It has become increasingly clear that insulin resistance and diabetogenesis are not merely a function of genetics and Western diet and lifestyle choices, but that environmental pollutants, such as persistent organic pollutants, pesticides, bisphenol A, and phthalates, are also considered risk factors. These observations deserve further consideration in the context of our understanding that more than one estrogen receptor participates in estrogen action.

A role for GPER in estrogen-induced carcinogenesis is logically argued from the perspective that views “cancer as the chronic wound that does not heal” [113], and this idea has been reviewed previously [30]. In this context, GPER drives estrogen-induced carcinogenesis by triggering intracellular signaling pathways that allow for (i) a loss of epithelial cell polarity and invasion of the basement membrane associated with epithelial-to-mesenchymal transition (EMT) [114]; (ii) the invasion of local tissue via the activation of matrix metalloproteinases (MMPs) and other cell surface proteinases [115,116,117]; (iii) escape from apoptosis [118,119] and survival outside the confines of its basement membrane; (iv) the synthesis of a provisional fibronectin matrix [44,120]; (v) the production and release of soluble factors that alter local and peripheral immunity [35,121]; (vi) a mechanism to activate its signaling in a hypoxic environment [119,122,123] and to activate cancer-associated fibroblasts [124] that lead to the development of a reactive cancer stroma; and (vii) the elicitation of signals that promote the recruitment of a vascular supply to reoxygenate the “wounded” tissue [125], which ultimately may result in systemic tumor cell dissemination. GPER-mediated cellular responses associated with cancer are not limited to endogenous estrogens. The environmental pollutant BPA is linked to GPER action in testicular seminoma cells [126,127] and upregulates GPER target genes: c-fos, early growth response-1 protein (EGR-1), and connective tissue growth factor (CTGF) in breast cancer cells and stromal fibroblasts [128]. BPA may indirectly promote carcinogenesis through its diabetogenic effects as its administration results in hyperinsulinemia in mice [129] and humans [130]. More recent studies have shown that BPA is able to act via GPER to reverse the antiapoptotic effects of 17β-E2-mediated ER in insulin-producing pancreatic beta cells [128].

Despite the connection between GPER, carcinogenesis, and cancer, it is surprising that antiestrogen therapies for breast cancer remain persistently designed to target only ER. Moreover, the fact that GPER directly associates with tumor size and distant metastasis [56,58,59], while ER varies inversely with these disease progression variables [131], suggests that ER and GPER may collaboratively and complementarily contribute to breast carcinogenesis [132]. This hypothesis is strongly supported by a retrospective clinical study of 742 ER+ low-risk (lymph node-negative, postmenopausal) breast cancer patients whose primary tumors were GPER-negative and showed improved disease-free survival [57]. Experimental data also support a pro-oncogenic role for GPER in breast cancer. Firstly, it has been shown that GPER is required for the survival of tumor-initiating “stem” cells derived from patient-derived tumor xenografts (PDXs) [60]. Secondly, less aggressive disease occurs upon the somatic cell silencing of GPER in a robust model of PyMT-induced breast carcinogenesis [133]. Additionally, GPER-selective ligands have shown efficacy in nonclinical and clinical settings. For example, the GPER antagonist, G36, delays the estrogen-dependent outgrowth of transplanted ER−/GPER+ endometrial carcinoma in mice [65], while the GPER agonist, G1, has been shown to attenuate the growth of patient-derived xenografts of pancreatic ductal adenocarcinoma [77]. Lastly, it is also important to consider the fact that certain ER antagonists such as tamoxifen, fulvestrant, and raloxifene act as GPER agonists [6], with evidence that tamoxifen can upregulate aromatase activity in tumor and stromal fibroblasts [134]. Thus, the broad role GPER plays in carcinogenesis calls into question the extent to which the current rubric for endocrine therapy can truly be effective beyond tumors that are only ER involved—in other words, for cancers such as ER+/GPER+ and ER−/GPER+, which account for almost half of all breast cancer types (discussed in more detail in Section 10 below) [56,135]. Collectively, these observations strongly suggest a link between the effectiveness of cancer therapy and the involvement of GPER.

## 7. GPER-Targeted Drugs for Metaboregulation and Cancer

GPCR-targeted drugs account for approximately 35% of all FDA-approved drugs [136]. Collectively, these drugs serve as effective treatments for modulating a wide assortment of diseases affecting almost every known physiological system. Experimental data strongly support the idea that GPCRs play a critical role in cancer, which is consistent with their role in tissue homeostasis, and this has been reviewed in detail [137,138,139,140,141]. In brief, these data include (i) the demonstration of a link between excessive signaling by GPCR cancers such as Mas-1, angiotensin II metabolite, angiotensin-(1-7) neuropeptides, gut hormones, and neuroendocrine and digestive cancers; (ii) findings showing that the constitutively acting mutations (CAMs) of GPCRs are associated with head and neck squamous cell carcinoma, small-cell lung cancer, thyroid cancer, pancreatic cancer, and prostate cancer; and (iii) laboratory evidence pointing to the overexpression of the α1B adrenoceptor (α1B-AR) in rodent fibroblasts resulted in focus formation. Further supporting a case for dysregulated GPCR signaling and cancer is the concept that ligands that act through GPCRs promote cellular responses associated with a loss of tissue homeostasis. Chemokine receptors are well known for their central role in leukocyte diapedesis, the process through which immune cells extravasate from the vasculature and invade parenchymal tissue. Moreover, smoothened receptors and frizzled receptors, as well as their Wnt ligands and hedgehog ligands, play critical roles in development and embryogenesis and have attracted significant interest as therapeutic targets for cancer [142,143,144,145]. Finally, it is important to recognize that targeting GPCRs is a successful strategy for the treatment of breast cancer, as the disruption of the neuroendocrine–ovarian cascade of estrogen biosynthesis, leading to ovarian suppression, is effectively achieved through the bolus administration of LHRH superagonists [146].

From the prevalence of GPCRs in current drug design and the role they play in cancer, it is not surprising that the development of cancer drugs that target GPCRs is gaining traction [141,147,148,149]. A variety of different treatment strategies are under development, including small molecule pharmacological compounds that target GPCRs and G proteins [150], as well as antibody-based approaches [84]. In 2006, Prossnitz and colleagues developed a small molecule selective GPER agonist, named G-1 [151], that has been instrumental in deciphering the individual molecular signaling activities of GPER, ERα, and ERβ. Since its discovery, G-1 has been used by a large number of investigators to assess the pathophysiological role of GPER in various disease states, including diabetes and obesity [109,152], autoimmune encephalomyelitis [152], hypertension [153], atherosclerosis [154], endometriosis [27], rheumatoid arthritis [155], acute colitis [156], and schizophrenia [157]. These studies have not only demonstrated the involvement of GPER in these estrogen-induced pathological conditions, but they have also served to demonstrate that G-1 is a relatively safe compound that does not elicit overt toxicological responses in mice, even when used in the long term. Because of the link between metabolic disease and cancer, research showing that G-1 (under preclinical development as Tespria™) can reverse obesity and improve glucose homeostasis in mice that are fed an obesogenic diet may be particularly relevant [109]. As was discussed earlier, Ridky and colleagues have shown that GPER agonism can also heighten PD-L1 sensitivity in a preclinical model of pancreatic ductal adenocarcinoma [77]. In another study, the stimulation of uveal melanoma cells with LNS8801 also induced melanocyte differentiation markers and concomitantly upregulated p53 and p21, G_2_/M mitotic arrest and the disruption of the mitotic spindle [63]. Whether or not these results are associated with cAMP-dependent GPER signaling or receptor downmodulation is unclear at present. Similarly, G2/M arrest was achieved in GPER-positive human HCC1806 TNBC cells following treatment with a GPER-targeted proteolytic targeting chimera, while cell cycle blockade did not occur in GPER-low human MDA-MB-231 breast cancer cells [78]. Recently reported Phase 1 results demonstrated that LNS8801 alone and in combination with pembrolizumab (Keytruda^®^) was tolerable and showed antitumor activity in 15 patients with metastatic uveal melanoma (mUM) studied to date. The fact that the mUM cohort from this study featured an overrepresented hypofunctional germline GPER variant compared with the normal population suggested a potentially significant role for GPER in the development of therapies to treat this for melanoma [158,159].

An alternative method for targeting GPER has been demonstrated by the development of GPER antagonists. Several have been developed, including G15, G36, CIMBA, MIBE, PBX1, PBX2, C4PY, and CPT, with half-inhibitory constants (IC_50_) that range from 200 to 5000 nM [160,161,162,163]. Among these GPER antagonists, G15 is noteworthy due to its high binding affinity for GPER (K_d_ = 20 nM) [161]. While the ERα affinity for G15 is quite low (>10 mM), an improved GPER antagonist, named G36, was developed by the same investigators with still lower affinity for ERα, and the ability to block the growth of transplanted ER-negative type II endometrial cancer cells [65]. The GPER antagonists G15/G36 have been developed based on their ability to prevent G-1 binding to GPER and inhibit GPER-dependent 17β-E2 signaling (K_i_ = 200 nM) [161,164]. An independent series of GPER antagonists with an affinity for GPER and in vivo efficacy was developed by Arnatt and coworkers. The lead compound in this series is, 2-cyclohexyl-4-isopropyl-N-(4-methoxybenzyl) aniline (CIMBA), which shows good selectivity and inhibitory activity (IC_50_ = 75 nM) and inhibits the formation of estrogen-induced cholesterol gallstones in mice [160]. Whether the mechanism associated with efficacy is due to pharmacological activation, the inhibition of GPER and/or its downmodulation has yet to be clearly determined. Other modalities of inhibiting GPER, including targeting G proteins, downstream effectors of GPER, and the use of therapeutic antibodies and antibody conjugates, have been discussed previously [84]. 

## 8. Rationale for Assignment of Endocrine Therapy for Breast Cancer

Guidelines for Estrogen and Progesterone Receptor Testing in Breast Cancer recommend the measurement of ER in biopsied breast tumor tissue using a standardized, semiquantitative immunohistochemistry test that is used to diagnose cancer as the rationale basis for predicting which patients may benefit from endocrine therapy (although in practice, clinicians may prescribe treatment regimens that include endocrine therapy to breast cancer patients when it is not clear that it will be efficacious). At present, no other assays are recommended for this purpose. However, there is a large variation in the presence or absence of ER that helps determine the eligibility for endocrine treatment. This is underscored by the findings of the 2020 update guideline from the American Society of Clinical Oncology (ASCO)/College of American Pathologists (CAP) that acknowledged that limited data have shown the benefit of endocrine therapy for tumors with ER 1–10% expression, which were termed “ER-low positive breast cancer” [165,166]. Moreover, the current standardized test for the presence of ER in tumor tissue does not discriminate whether biopsied tumor tissue expresses ERα, ERβ, or both. Significantly, GPER expression is not currently evaluated in tumor biopsy specimens, despite its established role in breast cancer carcinogenesis, and corroborating data from a significant number of studies have shown that GPER is expressed independently from ER in treatment-naïve breast tumors [49,56,57,167,168].

The consequence of reliance on ER as the canonical breast cancer-involved estrogen receptor is that therapies available to date and those under development are designed to target and treat this receptor and not GPER. Thus, patients receiving endocrine therapy to treat tumors that express both ER and GPER are at risk of being inadequately treated, as shown in Figure 3 [56,135]. 

The finding that GPER and ER are independently expressed in breast cancer suggests that ER-only-targeted endocrine treatments, including those under development, adequately treat only 19% of breast cancer patients (ER+/GPER−, red-shaded area), while 43% of all patients (ER+/GPER+, blue-shaded area) are treated partially. The ER−/GPER+ and ER−/GPER− expressed breast cancer (green- and grey-shaded areas, respectively) account for 19% each and are not expected to respond to ER-based treatment. Perhaps most significantly, unlike ER, GPER is expressed in the majority of triple-negative breast cancer (TNBC), providing a potential therapeutic target for this aggressive breast cancer subtype [50,51,52]. 

## 9. Evaluation of FDA-Approved Endocrine Therapies for the Treatment of Breast Cancer

As has been discussed, the onset and proliferation of estrogen-related breast cancer are controlled by the prevalence of, and interplay between, estrogen or estrogen-like (environmental) ligands and estrogen receptors (ER and GPER) that results in the formation of estrogen–estrogen receptor complexes and subsequent signaling that promotes carcinogenesis. Clinically, however, the drug development of endocrine therapy for the treatment of breast cancer has focused on addressing two complementary approaches to prevent the formation of an estrogen–estrogen receptor complex—i.e., 17β-E2–ER complex—by using small molecule drugs that either (i) downregulate estrogen production within the body or (ii) downregulate ER function. The former approach targets the machinery that produces estrogen in the body and includes the use of aromatase inhibitors (AIs) and luteinizing hormone-releasing hormone (LHRH) analogs (also called GnRH receptor modulators). AIs are molecules that bind either reversibly or irreversibly to the enzyme aromatase and prevent the conversion of androgens such as testosterone to estrogen, estradiol (17β-E2) [169], the penultimate step in estrogen biosynthesis. LHRH analogs suppress the secretion of gonadotropin, luteinizing hormone (LH), and follicle-stimulating hormone (FSH), resulting in a reduction in estrogen production in the ovaries [170].

In contrast to controlling the production of estrogen in the body, the complementary approach to endocrine therapy focuses on the estrogen receptors themselves by utilizing ligand molecules that can bind to ER competitively with estrogen, thus preventing the formation of the 17β-E2–ER complex required to carry out signaling and transcriptional functions necessary for cancer cell growth. Drug development focused on the downregulation of estrogen receptor function has further evolved into two approaches: (i) selective estrogen receptor modulators (SERMs) that bind to estrogen receptors as regulators/inhibitors, and (ii) selective estrogen receptor degraders (SERDs) that bind to and subsequently induce the destruction of estrogen receptor proteins via innate ubiquitination processes responsible for maintaining tissue homeostasis. Since its inception nearly 60 years ago, endocrine therapy has evolved, and advances have been made on two fronts. First, the combinatorial use of more than one treatment modality to reduce endogenous estrogen and/or block ER signaling (i.e., pairing ovarian suppression with an aromatase inhibitor or an aromatase inhibitor with a SERM such as tamoxifen) has been shown to be slightly more effective in treating premenopausal women [146]. The second area of advancement has been in developing SERDs with better bioavailability (e.g., elacestrant and camizestrant), although it remains to be seen whether these agents are significantly different than fulvestrant, discussed in a later section. A summary of historical and current approaches to endocrine therapy is discussed below and evaluated in terms of their ability to account for key elements that contribute to breast cancer (Table 2).

### 9.1. Aromatase Inhibitors (AIs)

Aromatase inhibitors were first developed in the early 1960s, leading to the development and European Medicines Agency (EMA) approval of formestane (marketed as Lentaron™ by Lisboa B.P.) in 1992, indicated as a first-line treatment for estrogen-receptor-positive (ER+) breast cancer in postmenopausal women. Formestane is steroidal-based, administered intramuscularly, and is currently only sparsely used in the EU, South America, and Asia [171,172]. Since then, other steroidal and nonsteroidal drugs have been developed to enable oral administration, leading to the approval of anastrozole (Arimidex™, nonsteroidal), exemestane (Aromasin™, steroidal), and letrozole (Femara™, nonsteroidal) that are used today. AIs are generally used as a first-line therapy in adjuvant and metastatic settings to treat postmenopausal women with hormone-responsive cancer [170,173] and have been shown to reduce the risk for recurrence postcurative treatment. In the adjuvant setting, AIs are often used in combination with other endocrine therapies such as LHRH inhibitors and tamoxifen (a SERM, discussed below), and for metastatic disease, they are often prescribed with a CDK4/6 inhibitor [174]. Common side effects from AIs include symptoms of menopause (hot flashes, night sweats, and vaginal dryness), joint and muscle pain, bone brittleness and loss, and raised levels of cholesterol [175].

AIs are ineffective in premenopausal women where estrogen production also occurs in the ovaries. However, they may be utilized when combined with ovarian suppression via either the surgical resection of the ovaries (oophorectomy) or the use of a drug such as leuprolide (Lupron™) or goserelin (Zoladex™) [176,177]. For example, a recently published meta-analysis of premenopausal women (7030) with ER+ breast cancer demonstrated that the use of AIs in conjunction with an ovarian suppressor such as an LRHR agonist (below) or ablation reduced the risk of breast cancer recurrence versus the SERM tamoxifen [178]. While AIs have shown promise in endocrine therapy by blocking the interconversion of androgens to estrogens, they obviously have no effect on the biological actions of a multitude of environmental estrogens, including phytoestrogens as well as estrogen-like molecules known to function as EDCs. Such chemicals are found in plasticizers, household products, cosmetics, and organopesticides such as BPA, which has been extensively used in plastics and is a potent GPER agonist [100].

### 9.2. Luteinizing Hormone-Releasing Hormone (LHRH) Analogs

LHRH analogs, also referred to as gonadotropin-releasing hormone modulators (GnRH modulators), are agonists or antagonists that modulate the GnRH receptor. LHRH analogs can enable the elevation or suppression of hormone secretions within the gonads and subsequently increase or reduce hormone levels (estradiol, progesterone, and testosterone) [179]. In the context of breast cancer, LHRH agonists act as ovarian suppressors and include leuprolide (Lupron™) and goserelin (Zoladex™) [175,176,177]. They are effective for the treatment of disease in premenopausal women and are used alone or in combination with AIs, SERMs, and SERDs (below) in an adjuvant or metastatic setting [170,176,177,180]. Common side effects are typical of menopausal symptoms and include hot flashes, vaginal dryness, and mood swings [175]. However, like AIs, these pharmacological agents are limited in their ability to block neuroendocrine-mediated estrogen biosynthesis through the hypothalamic–pituitary–ovarian axis. LHRH/GnRH drugs do not block nonovarian estrogen biosynthesis that occurs in a variety of tissues in postmenopausal women; nor do they have any effect on the activity of environmental estrogens.

### 9.3. Selective Estrogen Receptor Modulators (SERMs)

SERMs were first developed in the 1960s and 1970s for the treatment of infertility and/or menopause (clomifene, Clomid™, 1967; cyclofenil, Sexovid™, 1970). Since then, the development of SERMs continued with the discovery and approval of drugs for hormonal contraception (ormeloxifene, Saheli™, 1991) and osteoporosis (raloxifene, Evista™, 1997; lasofoxifene Fablyn™, 2009). In the context of breast cancer treatment, the most used SERM is tamoxifen (marketed as Nolvadex^®^, Genox^®^, and Tamifen^®^, among others), which was discovered in 1962, approved in 1978, and remains a first-line therapeutic mainstay to the extent of nearly 1 million prescriptions as of 2020 [181]. Tamoxifen is indicated for premenopausal and postmenopausal women and men and is typically administered orally for a period of 5 years, after surgery [182,183]. More recently, the National Comprehensive Cancer Network (NCCN) has recommended that the adjuvant administration of tamoxifen be extended to 10 years for patients with early-stage disease, based on the results from the ATLAS clinical trial [184], as well as for patients with metastatic disease [185]. For patients who are prescribed chemotherapy and endocrine therapy post surgery, tamoxifen typically follows chemotherapy. Since the advent of tamoxifen, other SERMs that have been developed for the treatment of breast cancer include raloxifene (Evista™, 1997) and toremifene (Fareston™, 1997) [186]. Compared with tamoxifen, raloxifene and toremifene have similar efficacy but also reduce the risks of other complications such as thromboembolic events and cataracts [187,188]. However, while raloxifene was approved for the prevention of breast cancer and has been shown to reduce the risk associated with endometrial cancer, this comes with an increased risk of DCIS in early-stage breast cancer [188]. 

As with other SERMs, tamoxifen acts as an allosteric inhibitor by competitively binding to estrogen receptors over estrogen, thus preventing the alpha–beta heterodimerization of the ER and the recognition of the ER–ERE gene promoter responsible for transcription factors necessary for cancer cell growth [189]. Nonserious side effects common to tamoxifen and other SERMs include irregular menstruation, weight loss, and hot flashes, with more serious side effects of a slightly increased risk of uterine cancer, stroke, vision problems, and pulmonary embolism [183]. Although SERMs such as tamoxifen have substantial benefit to patients and continue to be used in the treatment of breast cancer, they suffer from drug resistance resulting from the hypersensitivity of ERα to estrogen and mutations to the ligand binding domain or the stabilization of ERα that are either ligand-independent or SERM-induced [190,191]. Furthermore, it has been shown that tamoxifen and raloxifene act as GPER agonists_,_ raising the possibility that whatever efficacy they provide in silencing cancer growth signaling by antagonizing ER may be countered to some extent by promoting signaling via GPER. In addition, it has also been observed that while tamoxifen acts as an ER+ antagonist, it also acts as an agonist in the uterus, which can lead to an increased chance of endometrial cancer [192,193].

### 9.4. Selective Estrogen Receptor Degraders (SERDs)

ER signaling that is required for the proliferation of many breast cancers eventually becomes resistant to the positive, inhibitory effects of aromatase inhibitors or SERMs such as tamoxifen, due in part to the effect of mutations on the ER structure. Therefore, therapeutic approaches that can permanently disrupt signaling and transcriptional pathways represent a means of potentially thwarting drug resistance [194,195]. In contrast to aromatase inhibitors, LHRH agonists and antagonists such as SERMS, protein destabilization and degradation represent a third mechanism for downregulating the function of estrogen receptors [196]. Selective estrogen receptor degraders (or downregulators), SERDs, are a class of small molecules that principally function by binding to the ER estrogen receptor to form an ER–SERD complex with reduced mobility and stability, resulting in a conformational change in the protein and subsequent degradation of the receptor [197] and blocking transcriptional activity [198,199]. Thus, with respect to their interaction with estrogen receptors, SERDs are often described as functioning as event-driven drugs, whereas SERMs act as occupancy-driven drugs [200]. In addition to effecting degradation, SERDs can act antagonistically, somewhat analogous to SERMs, where the compromised mobility and altered conformation of the ER–SERD complex reduces the rate of intracellular nuclear translocation and the turnover of estrogen receptors and downregulates the transcription of ER-modulated genes [174]. By degrading, rather than simply inhibiting, receptor estrogen function, SERDs overcome the limitations of SERMS irreversibly reducing selective pressure from downmodulation that otherwise promotes mutations leading to drug resistance [201].

Fulvestrant, an injectable steroidal drug marketed by AstraZeneca as Faslodex™, is a first-generation SERD, and until very recently, it was the only FDA-approved example of such. It is indicated as a first- or second-line endocrine therapy for postmenopausal women with ER-positive metastatic breast cancer or locally advanced disease that is unresectable [202]. The mechanism of action for fulvestrant involves binding to the ER to form a fulvestrant–ER complex that prevents the dimerization and localization of ER in the nucleus, rendering it unstable toward degradation [195,203]. Fulvestrant has also shown efficacy toward ESR1 mutations in the ER, although the maintenance of clinical effectiveness requires elevated dosing [203,204]. Fulvestrant suffers from several limitations typical of SERDs, including eventual drug resistance and the need for painful, regular intramuscular administration due to poor oral bioavailability [193]. The sterically encumbered steroidal nature of fulvestrant also presents challenges in terms of chemical derivatization to improve bioavailability without compromising efficacy [205]. There are many SERDs in various stages of clinical development with the goal of overcoming such limitations. One such example is the drug elacestrant (marketed by the Menarini Group as Orserdu™), an orally administered, nonsteroidal hybrid SERM–SERD drug that was approved by the FDA in early 2023. It is indicated for the treatment of postmenopausal women or adult men with ER-positive, HER2-negative, ESR1 mutated advanced or metastatic breast cancer with disease progression following at least one line of endocrine therapy [206]. From Phase 3 clinical studies (the EMERALD study), elacestrant showed improved progression-free survival for patients with ESR1 mutations (which are largely responsible for drug resistance) and exhibited manageable safety for patients with ER-positive/HER2-negative advanced breast cancer versus standard of care (SOC) fulvestrant or AIs [207]. Another example of a second-generation SERD targeting improved bioavailability is camizestrant, under development by AstraZeneca. The results from a recent Phase 2 study (SERENA-2) demonstrated potential superiority over fulvestrant in patients with HER2-negative breast cancer, including a two-fold increase in progression-free survival attributed to a reduction in the levels of ESR1 mutation [208,209]. As was noted for tamoxifen, fulvestrant and raloxifene can also act as GPER agonists [191], thus creating a situation where their overall efficacy is likely predicated on their antagonism of ER overriding their agonism of GPER. It remains to be seen whether elacestrant and camizestrant also demonstrate GPER agonism, and if so, the extent to which this may limit their efficacy as well.

## 10. Assessment of Approved Endocrine Therapies in View of the Current Rubric and the Effect of Elements Not Considered

Regulatory-approved endocrine therapeutics based on the current rubric have persistently focused on downregulating endogenous estrogen (17β-E2) production and ER signaling function and have been successful as therapies (Figure 4a). However, the completeness of their efficacy in terms of addressable cancer types, drug resistance, and sources of estrogen is ultimately limited by environmental estrogen and GPER elements not considered, along with the potential for the formation of a significant number of cancer cell growth-promoting complexes (Figure 4b; Table 2). Consequently, and as shown in Table 2, controlling endogenous estrogen production either by AI or LHRH analogs cannot account for the potential effects of environmental estrogens. Similarly, drug approaches that attempt to downregulate ER function and discount GPER are limited to cancers that are ER+ in nature and are not expected to be significantly efficacious for cancers such as TNBC, which has been shown to be GPER-regulated and not ER-regulated [50]. Moreover, therapies designed to target ER are expected to be most effective for ER+/GPER− cancers, whereas for ER+/GPER+ cancers, efficacy will always be limited by the extent of undrugged GPER involvement, or where such therapies adventitiously target GPER. The observation that drugs designed to downregulate ER function (ER drugs, Figure 4b) can act as agonists further confounds the picture for endocrine therapy. Clearly, the current rubric for endocrine therapy must therefore be reconsidered to account for all endocrine elements involved in promoting cancer cell growth. In particular, the ability to effectively mitigate the effect of environmental estrogens rests on the downmodulation of ER and GPER.

## 11. New Approaches to Targeted Protein Degradation and Future Considerations

Protein degradation approaches in endocrine therapy have emerged as a necessary element in front-line endocrine therapy by offering a means to account for not only the irreversible silencing of ER but also the potential effects of environmental estrogens that can agonize this receptor. One of the principal limitations of degraders such as SERDs, e.g., fulvestrant, is that their efficacy is tied to their ability to bind to the ligand binding pocket (LBP) of ER to form a SERD–ER complex (notwithstanding the role GPER plays in carcinogenesis and the need to target this critical receptor in addition to ER). Mutations to the ER can therefore lead to changes in the structure of the LBP and the inability of the SERD to properly bind, leading to drug resistance. Moreover, the degradation of the SERD–ER complex via tissue homeostasis relies on the efficiency of the body’s innate ubiquitination processes to recognize and target the affected ER. To overcome such limitations, rational drug design for endocrine therapy has evolved into a variety of novel targeted protein degradation strategies, which include molecular glues, lysosome-targeting chimeras (LYTACs), antibody-targeting chimeras (AbTACs), and proteolysis-targeting chimeras (PROTACs) [210].

PROTACs have received considerable attention in the past several decades since their initial discovery by Sakamoto and Crews [211] and are the subject of numerous clinical trials, notably including the PROTAC ARV-471 currently being studied in Phase 3 clinical trials for patients with ER+/HER2- breast cancer. PROTACs are heterobifunctional protein degrader molecules that comprise two protein-binding ligands and a flexible linker that connects them. One of the ligands is designed to bind to the protein of interest (POI) targeted for degradation, while the other ligand is designed to bind to the E3 ligand ligase protein responsible for ubiquitination and degradation. The mechanism of action involves the binding of the ligands of the PROTAC to their respective protein targets to form a ternary complex characterized by the simultaneous antagonism of the POI and the activation of the E3 ligase, inducing the direct degradation of the POI via ubiquitination. Following the degradation of the POI, the PROTAC is recycled and can degrade additional POIs [212]. PROTACs do not have to bind strongly to their POI or bind to their LBP to be effective. Therefore, they have the potential to be structurally modified to enhance solubility without affecting ligand binding—an aspect that is advantageous over SERDs. Furthermore, the ability for PROTACs to be recycled after each POI degradation event classifies them as catalytic rather than stoichiometric degraders. The catalytic nature of PROTACs therefore offers the potential for efficacy at substantially lower doses and within solubility limitations imposed by their structures compared with other endocrine therapy drugs [197,205].

Emergent protein degradation strategies such as PROTACs are exciting and offer significant promise in the treatment of breast cancer and address some of the shortcomings of SERDs. However, such strategies can only be most effective if their designs account for the mitigation of all estrogen and estrogen receptor interactions responsible for cancer incidence and proliferation.

## 12. Conclusions and Outlook

Endocrine therapy is the most widely prescribed and efficacious cancer therapy. Single orally delivered agents, including AIs and SERMs, are the cornerstone of front-line therapy for early-stage, ER+ breast cancer in postmenopausal women. Over the past several years, new methods of inhibiting estrogen action have been incorporated into the treatment of ER+ breast cancer, including the use of pharmaceutical agents that block ovarian suppression and the development of ER degraders such as SERDs with improved bioavailability. However, despite these improvements, antiestrogens require prolonged use, resulting in adverse effects, including osteoporosis, an increased risk of endometrial cancer and thrombosis, and ultimately drug resistance [213]. These undesirable consequences have in turn led to complex explanations for endocrine resistance while simultaneously ignoring the role played by environmental estrogens and GPER in the genesis of breast cancer, as well as a variety of other malignancies. There is a significant need to improve endocrine therapy to address the deficiencies in current modalities in terms of (i) accounting for the effects of all estrogens, (ii) targeting all cancer-involved estrogen receptors, and (iii) accounting for antagonizing one estrogen receptor target while unwittingly agonizing the another.

The global rise in cancer incidence, the known carcinogenic influence of environmental estrogens, their abundance in economically developed countries, and the prominent role that GPER plays in metabolic syndrome, a cancer-promoting health condition, are additional reasons for developing GPER-targeted cancer therapeutics. Given the fact that AIs and ovarian suppression are ineffectual to the carcinogenic potential of environmental estrogens, the effective mitigation of the role played by all estrogens in cancer proliferation will rely on the targeted protein degradation of nuclear estrogen receptors and GPER. Presently, GPER remains an “undrugged” therapeutic target in this space with tremendous and exciting potential to enable a more complete approach to endocrine therapy for breast cancer and other solid malignancies. Clinical and experimental data strongly support the need for developing GPER-targeted therapies. These findings include a direct association of GPER with clinicopathological indices that predict advanced disease, as measured by stage progression, poor patient outcomes, and endocrine resistance. Significantly, GPER expression occurs commonly in ER breast cancer and TNBC—breast cancer subtypes that are not considered candidates for the rationale assignment of endocrine therapy [49,50,135]. In preclinical studies, the genetic silencing of GPER in breast cancer resulted in significantly smaller tumors and reduced cancer stem cell activity [60,133]. Similarly, the pharmacologic modulation of GPER resulted in tumor suppression in preclinical models of endometrial [65], nonsmall cell lung carcinoma [77], pancreatic adenocarcinoma [66], and melanoma [64].

Significantly, Linnaeus Therapeutics has made progress toward the development of drugs targeting GPER. Clinical trial NCT 04130516, a Phase 1/2, first-in-human, open-label, multicenter study is underway to establish the safety, tolerability, PK, and antitumor effects of the GPER agonist, LNS8801, alone and in combination with the checkpoint inhibitor pembrolizumab. Early clinical results show favorable toxicology and a partial antitumor response in one patient [158]. Future results from the dose escalation phase of this study, as well as future clinical trials assessing the value of GPER as a therapeutic target, will help to establish the value of this promising therapeutic target. Successful results from these types of clinical studies have the potential to help expand our current view of which patients may benefit from antiestrogen therapy and should complement existing modalities for antiestrogen therapy.

## Figures and Tables

**Figure 1 cells-12-02460-f001:**
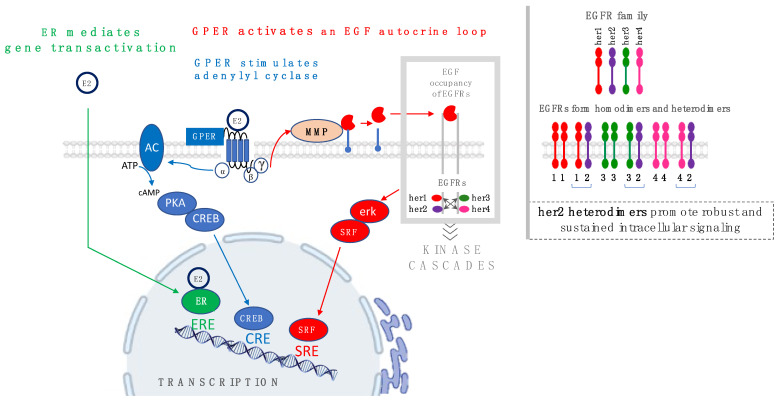
Schematic diagram showing the receptor-mediated signaling mechanisms that promote estrogen-regulated gene transcription: ER mediates direct gene transactivation by binding to estrogen-regulated enhancers (EREs) (green arrows). In contrast, GPER promotes gene transactivation indirectly as a downstream consequence of parallel cAMP-dependent (blue arrows) and EGFR-dependent (red arrows) signaling pathways that may be regulated by cyclic AMP-regulated enhancers (CREs) or serum-regulated enhancers (SREs). The inset in the upper right shows that members of the EGFR family form homodimers and heterodimers, and that prolonged and robust EGFR signaling occurs in cells in which her2/neu is amplified.

**Figure 2 cells-12-02460-f002:**
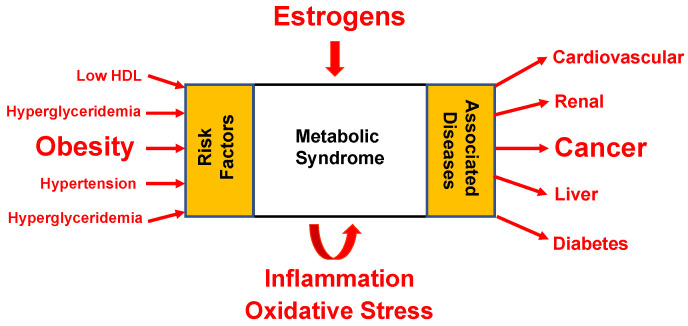
Schematic showing the relationship between estrogens, metabolic syndrome, and cancer. The hallmark features of metabolic syndrome are clustered around abdominal obesity and are measured as hypertension, hyperglycemia, hyperglyceridemia, and low serum concentrations of high-density lipoproteins. Individuals with metabolic syndrome are likely to develop several associated diseases, including cardiovascular, renal, and liver disease, as well as diabetes and cancer. In the model above, estrogens (both environmental and endogenous) act as toxicants, which further contribute to disease progression (adapted from Vijayalakshmi Varma, National Center for Toxicology Studies, USDA) [111].

**Figure 3 cells-12-02460-f003:**
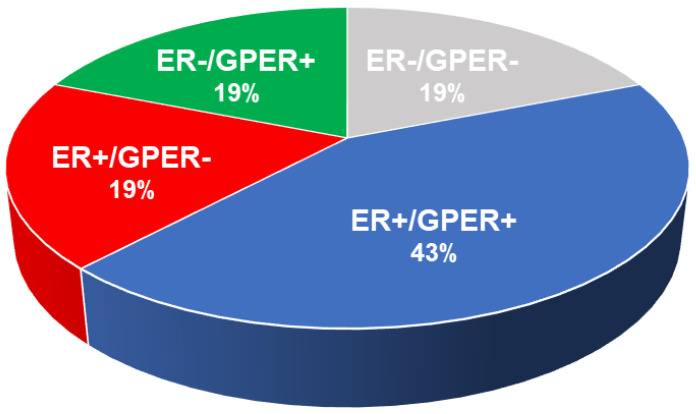
Distribution of breast cancer by the presence of ER and GPER in biopsy specimens [56,135].

**Figure 4 cells-12-02460-f004:**
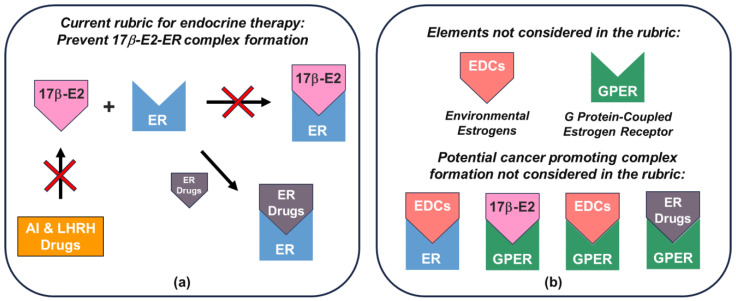
Schematic depicting (**a**) the current rubric for rational assignment of endocrine therapy and (**b**) elements not considered in the rubric. Panel (**a**) shows current endocrine therapies that prevent the formation of endogenous estrogen (17β-E2)–ER complex by either blocking endogenous estrogen biosynthesis (via aromatase inhibitors or LHRH agonists, orange box) or through the molecular action of estrogen at the estrogen receptor ER via ER Drugs (i.e., SERMs/SERDs, grey icon). Panel (**b**) also shows elements and interactions that are not accounted for in the current rubric: (i) endocrine-disrupting chemicals in the environment (EDCs, i.e., xenoestrogens and phytoestrogens; red icon) and GPER (green icon), and (ii) cancer-promoting complexes resulting from combinations of 17β-E2 and EDCs that can interact with and agonize ER and GPER, and ER drugs (antagonists) that can also interact with and function as GPER agonists.

**Table 1 cells-12-02460-t001:** Relative binding affinities for 17**β**-E2 and examples of environmental estrogens to GPER, ER**α**, and ER**β** ^1^. “*” indicates a multiplication operator.

Estrogen	Structure	% Relative Binding Affinity: Affinity_Estrogen_/Affinity_17β-E2_ * 100%		
		ERα	ERβ	GPER ^5^
17β-E2	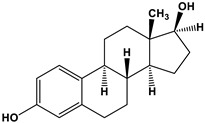	100	100	100
Bisphenol A	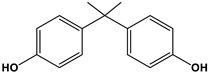	0.008 ^2^	0.008 ^2^	0.5–1
Genistein	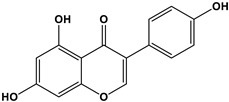	0.021 ^3^	6.8 ^3^	2–5
Daidzein	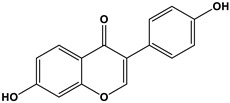	0.003 ^3^	0.051 ^3^	N/D
4-Hydroxytamoxifen	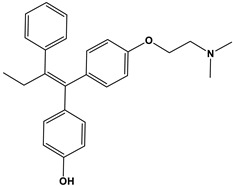	257 ^4^	232 ^4^	0.3–6

^1^ Except for 17**β**-E2, RBA ranges shown reflect ranges in affinity values originating from different sources/methods; ^2^ values reported for ER in [39]; no distinction between ERα or ER**β** was given; ^3^ values reported in [80]; ^4^ values reported in [81]. Note that Blair reported an RBA for 4-hydroxytamoxifen for GPER of 175% [39]; ^5^ GPER values calculated from data reported in [6].

**Table 2 cells-12-02460-t002:** Modalities and attributes of endocrine treatment for regulatory approved drugs.

Therapy/Attributes	AI	LHRH Analog	SERM	SERD
*Class*	Inhibitor	Inhibitor	Inhibitor	Degrader (noncatalytic)
*Receptor Target*	Aromatase	GnRH Receptor	ER	ER
*Downregulation MOA*	Inhibition of estrogen biosynthesis	Ovarian suppression	ER-mediated gene transactivation	ER degradation and signaling
*Examples*	Exemestane, Anastrozole, Letrozole	Leuprolide, Goserelin, Triptorelin	Tamoxifen, Toremifene, Raloxifene	Fulvestrant, Elacestrant
*Treatment*	First line	First line	First line	Second line
*ER*+/*GPER*+	Yes	Yes	Yes ^a^	Yes ^a^
*ER*+/*GPER*−	Yes	Yes	Yes	Yes
*ER*−/*GPER*+	No	No	No	No
*ER*−/*GPER*−	No	No	No	No
*TNBC*	No	No	No	No
*Drug Resistance*	Yes	Yes	Yes	Yes
*Endogenous Estrogen*	Yes	Yes	No	No
*Environmental Estrogen*	No	No	No	No
*Premenopausal*	Yes ^b^	Yes ^c^	Yes ^d^	No
*Postmenopausal*	Yes	Yes	Yes	Yes

^a^ Efficacy is expected to be predicated to the extent that ER involvement overrides GPER involvement in cancer cell growth and/or these drugs adventitiously inhibit or degrade GPER. ^b^ In combination with SERMS. ^c^ In combination with AIs and SERMS. ^d^ In combination with AIs.

## Data Availability

No new data are reported herein. The data presented in this article are openly available via internet sources and websites such as PubChem (https://pubchem.ncbi.nlm.nih.gov/, accessed on 3 October 2023).

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
