# Peer review of "The G Protein-Coupled Estrogen Receptor (GPER): A Critical Therapeutic Target for Cancer"

_cells, 2023, doi:10.3390/cells12202460_

Round 1
Reviewer 1 Report
In this review, the Authors provide an exhaustive and captivating overview on the potential of the G protein estrogen receptor as a valuable therapeutic target for cancer. In particular, the Authors focus on the estrogen receptor-based mechanisms by which endogenous and environmental estrogens exert a pro-tumorigenic role in human malignancies, paying specific attention to GPER as a therapeutic target in metabolic disease and cancer. Next, the most fascinating section of the manuscript is focused on certain elements that are not currently contemplated in the rubric for rational assignment of ET. This overview strongly supports the need to modify the current rubric for ET by accounting for all elements (i.e. the link between environmental estrogens with both the classical ERs and GPER as well as the GPER-mediated oncogenic effects of endogenous estrogens) implicated in cancer progression.
I would just suggest to revise the text since some symbols do not appear and inappropiate high dashes are present.
Author Response
We thank the reviewer for their thorough review of our manuscript and the complimentary remarks. It has been a pleasure to watch the field grow over the past twenty years, which is the result of hard work by many. To the best of our knowledge, we have addressed all formatting issues that were present throughout the original manuscript.
Reviewer 2 Report
In the manuscript, the authors present a comprehensive and engaging analysis of the potential of the G protein estrogen receptor as a promising therapeutic target for cancer. The authors of this study specifically concentrate on the mechanisms involving estrogen receptors through which both endogenous and environmental estrogens contribute to the development of cancer in humans. They give particular emphasis to GPER as a potential target for therapeutic interventions in metabolic disease and cancer. Subsequently, the manuscript delves into a captivating section that explores specific elements that are presently not taken into consideration in the framework for the logical allocation of extraterrestrial entities. This overview provides substantial evidence in favor of revising the existing rubric for endocrine therapy (ET) to incorporate all relevant factors, including the association between environmental estrogens and both the classical estrogen receptors (ERs) and G-protein coupled estrogen receptor (GPER), as well as the GPER-mediated oncogenic effects of endogenous estrogens, which are known to contribute to the advancement of cancer.
Overall, the paper is well written and the discussion of GPER as a therapeutic target in metabolic disease and cancer is interesting. Although the authors introduce some basic concepts in the section, I feel they are important and an addition to the existing literature. I recommend the paper for publication.
I ask the authors to address my small only concern below:
I found unnecessary punctuation “_” and spaces throughout the text.
Author Response

(The authors gave the same response as above.)

Reviewer 3 Report
The manuscript by Hall and colleagues is a review examining the therapeutically potential of G coupled estrogen receptors for cancer with respect of the classical approach based on estrogen hormonal therapy targeting nuclear estrogen receptors. The manuscript cover many aspect of estrogen signaling and review a broad range of FDA approved treatments. In this perspective the manuscript is complete and very informative. Authors have done a very great job. However, I noticed several issues that authors should correct.
1.In all the manuscript there are words cut in two piece separated by a “-“. See example line 27, 32 and 39. Please check and correct.
2.In the abstract, line 18, the sentence sound awkward. It seems a verb is missing.
3. I think the 1st sentence of the introduction is not necessary.
4. In many occurrences, some letters are missing such as Greek letter. See example line 91 and 93 to define endogenous estrogen. This is the same for the two estrogen receptors ER alpha and beta (line 97)
5. Please define HB-EGF
6. In Figure 1, some letters are too small. MMP is not define in the legend and in the text. What is the protein cleaved by MMP?
7. What is complement of estrogen receptors in line 166
8. Line 172 GPER ‘gene’ instead of GPER alone.
Author Response
We thank the reviewer for their thorough review of our manuscript and the complimentary remarks. It has been a pleasure to watch the field grow over the past twenty years, which is the result of hard work by many. We have provided answers in italics following your specific comments below.
- In all the manuscript there are words cut in two piece separated by a “-“. See example line 27, 32 and 39. Please check and correct. Thank you for pointing this out. This is a formatting issue that has been addressed.
- In the abstract, line 18, the sentence sound awkward. It seems a verb is missing. Thank you for pointing this out - it has been corrected.
- I think the 1st sentence of the introduction is not necessary. We respectfully disagree as it is important to demonstrate the scope of the problem and the rate at which it is increasing. No change was made here.
- In many occurrences, some letters are missing such as Greek letter. See example line 91 and 93 to define endogenous estrogen. This is the same for the two estrogen receptors ER alpha and beta (line 97). Thank you for pointing this out. This is a formatting issue that has been addressed.
- Please define HB-EGF. We have defined this acronym.
- In Figure 1, some letters are too small. MMP is not define in the legend and in the text. What is the protein cleaved by MMP? We have edited the figure which also now identifies the target of MMP as HB-EGF.
- What is complement of estrogen receptors in line 166. As used here, “complement” refers to a number or quantity of something, especially that required to make a group complete. For example, "at the moment we have a full complement of staff". In the context of the article, we are referring to ER and GPER as the complement of receptors.
- Line 172 GPER ‘gene’ instead of GPER alone. Thank you for pointing this out. This change has been made.
Reviewer 4 Report
The manuscript provides a comprehensive and up-to-date overview on the G-protein coupled estrogen receptor (GPER) as a potential therapeutic target in cancer. In particular, the mechanisms of action, agonist and antagonist molecules as well as the current knowledge regarding the clinical role of GPER have been discussed. Special emphasis has been given to the similarities and differences between GPER and the classical estrogen receptor (ER)α in diverse contexts. Interestingly, the manuscript encompasses a thorough examination of the FDA-approved endocrine therapies for breast cancer treatment, pointing out the need to consider not only the beneficial effects exerted through ERα, but also the potential contrasting outcomes elicited through GPER. The manuscript is well-structured and clearly written. The Authors have gathered and presented inclusive information, highlighting key advancements in the field. Hence, few suggestions are indicated from this Reviewer.
As the Authors reviewed literature data regarding the link among GPER, metabolic diseases and cancer, it could be interesting to include a more in-depth discussion about the crosstalk between ER/GPER and the Insulin/IGF pathway. In the framework of the findings concerning the potential implication of estrogen-mimetics in carcinogenesis, it would be useful to provide further information on the involvement of GPER in the actions elicited by BPA in the breast tumor microenvironment (PMID: 22552965). Moreover, the role of GPER in breast cancer-related angiogenesis should be mentioned with a view to a translational prospective (PMID:29212519; 24894716; 23947803).
Author Response
We thank the reviewer for their thorough review of our manuscript and the complimentary remarks. It has been a pleasure to watch the field grow over the past twenty years, which is the result of hard work by many. We have added a paragraph within section 6, "GPER as a therapeutic target in metabolic disease and cancer", that reviews the pro-oncogenic cellular changes that are mediated by GPER and have included many of the references that are cited by the reviewer as well as a few others. The paragraph of interest start at line 363 in the revised manuscript.